# Towards Standardisation of a Diffuse Midline Glioma Patient-Derived Xenograft Mouse Model Based on Suspension Matrices for Preclinical Research

**DOI:** 10.3390/biomedicines11020527

**Published:** 2023-02-11

**Authors:** Elvin ’t Hart, John Bianco, Helena C. Besse, Lois A. Chin Joe Kie, Lesley Cornet, Kimberly L. Eikelenboom, Thijs J.M. van den Broek, Marc Derieppe, Yan Su, Eelco W. Hoving, Mario G. Ries, Dannis G. van Vuurden

**Affiliations:** 1Princess Máxima Center for Pediatric Oncology, Heidelberglaan 25, 3584 CS Utrecht, The Netherlands; 2Center for Imaging Sciences, University Medical Center Utrecht, Heidelberglaan 100, 3584 CX Utrecht, The Netherlands

**Keywords:** HSJD-DIPG-007, diffuse midline glioma, PDX model, Matrigel, metastases

## Abstract

Diffuse midline glioma (DMG) is an aggressive brain tumour with high mortality and limited clinical therapeutic options. Although in vitro research has shown the effectiveness of medication, successful translation to the clinic remains elusive. A literature search highlighted the high variability and lack of standardisation in protocols applied for establishing the commonly used HSJD-DIPG-007 patient-derived xenograft (PDX) model, based on animal host, injection location, number of cells inoculated, volume, and suspension matrices. This study evaluated the HSJD-DIPG-007 PDX model with respect to its ability to mimic human disease progression for therapeutic testing in vivo. The mice received intracranial injections of HSJD-DIPG-007 cells suspended in either PBS or Matrigel. Survival, tumour growth, and metastases were assessed to evaluate differences in the suspension matrix used. After cell implantation, no severe side effects were observed. Additionally, no differences were detected in terms of survival or tumour growth between the two suspension groups. We observed delayed metastases in the Matrigel group, with a significant difference compared to mice with PBS-suspended cells. In conclusion, using Matrigel as a suspension matrix is a reliable method for establishing a DMG PDX mouse model, with delayed metastases formation and is a step forward to obtaining a standardised in vivo PDX model.

## 1. Introduction

Paediatric high-grade gliomas (pHGGs) are malignant brain tumours found in the hemispheres and midline structures of the brain and account for 10% of all central nervous system (CNS) tumours in children, while being responsible for 40% of all fatal cases. Diffuse intrinsic pontine glioma (DIPG) is a particularly aggressive and invasive pHGG subtype arising in the brainstem (pons) and has been recognised as a distinct type within the paediatric diffuse high-grade glioma family in the 5th edition of the WHO Classification of Tumours of the Central Nervous System [1], and as such, these tumours have been reclassified to ‘diffuse midline glioma, H3K27-altered’ (DMG). Alterations in H3K27 in DMG include point mutations at the histone H3K27M, predominantly with H3.3 expression and to a lesser degree H3.1 with up to 80% of tumours harbouring one of these mutations [2]. H3.3 mutations cause trimethylation loss of the chromatin with altered manifestations of the oncogenes and tumour-suppressor genes [3]. Loss of H3K27 trimethylation by overexpression of EZHIP has been observed in H3K27 wildtype DMG [4]. In addition, DMGs are also commonly associated with mutations in the TP53 gene (up to 60%) and to a lesser extent with mutations in PPM1D (up to 30%) [5]. Combined, these mutations increase the aggressiveness of DMGs and are associated with a poor overall prognosis. Genomic analyses have revealed that DMGs are molecularly complex, also harbouring mutations in ACVR1, ATRX, H3F3A, HIST1H3B/c, MYC, PDGFRA, PIK3CA, PTEN, and RB1 that can cooperate with mutated TP53 and PPM1D to promote tumour formation [2,6,7,8,9]. In addition to pontine localization, DMGs can occur in other midline structures, such as the thalamus and spinal cord.

Pontine DMGs are mainly diagnosed in children between 6 and 9 years of age. Rapid progression of this disease results in a median survival of 11 months and a 95% fatality rate within 2 years after diagnosis [10,11]. DMG is commonly found in the brainstem, a delicate brain region responsible for the execution of vital functions [1,12]. Clinical symptoms are caused by pressure of the tumour and dysfunction of the brainstem, resulting in cranial nerve deficits, such as facial and abducens nerve palsy, multiple cranial neuropathies, and long-tract and cerebellar signs, such as paresis and ataxia [13]. Because of the delicate location and invasive nature of DMG, radical surgery is impossible, while chemotherapy is complicated by the presence of the blood–brain barrier (BBB), preventing 98% and 100% of small and large molecules from entering the brain [14,15]. While tumour progression can cause BBB disruption and subsequently increased BBB permeability, most of the BBB in DMG remains intact over the course of the disease. Even when BBB disruption is observed, this occurs mostly at the core of the tumour lesion after onset of local tissue necrosis [16]. Therefore, the current standard of care of DMG is fractionated radiotherapy of 1.8–2 Gy daily cumulating to a total dose of 54–60 Gy, with concurrent temozolomide causing temporal tumour growth delay, but also inevitable recurrence [17]. Metastasis along the neuroaxis is rarely seen at diagnosis (2%) but can increase to up to 17.3% at disease progression [10,18], where an under-recognised pattern of subventricular spread was observed in the majority of investigated cases, with infiltration of the subventricular zone as well as tumour nodules in the frontal horns of the lateral ventricles [19].

Although intensive research has been conducted for the treatment of DMG, little clinical progress has been made to date [20]. Even though in vitro drug screening has evidenced several promising chemotherapeutic candidates for DMG treatment, the successful translation to preclinical in vivo studies has demonstrated to be challenging [21,22,23,24,25]. Additionally, therapeutic translational complexity is added due to the biological differences between patients and animal models of the disease [26]. Although small animals do not develop DMG spontaneously, in vivo studies are made possible by establishing genetically engineered mouse models (GEMMs) or patient-derived xenografts (PDXs) [27]. GEMM models have an altered genomic profile to mimic the human disease allowing genetic/fundamental research to be conducted, while PDX models use orthotopic injection of human primary DMG cells in (partially) immune-deficient animals. The role of these models in preclinical research is to facilitate recapitulation of human malignancies and the associated disease progression, allowing validation of therapeutic agents or interventional techniques before clinical trials [28,29].

HSJD-DIPG-007 is an established DMG cell line from the Sant Joan de Déu Hospital in Barcelona, derived from the autopsy of a radiotherapy-naive, 6-year-old male that died one month after diagnosis and received one course of chemotherapy (cisplatin and irinotecan). These HSJD-DIPG-007 tumour cells harbour mutations in H3F3A K27M, ACVR1 R206H, PPM1Dp.P428fs, and PIK3CAp.H1047R [5,30]. In recent years, HSJD-DIPG-007 has increasingly been used as a cell line for DMG PDX mouse models [31]. This model displays an intact BBB as well as an invasive growth pattern that mimics human pathology for a large part of disease progression, rendering it appealing for evaluating therapeutic response and efficiency [32]. However, a standardised method in establishing orthotopic in vivo models using HSJD-DIPG-007 cells has not yet been developed. Current protocols using this cell type vary between studies on several levels, such as use of cell suspension matrix, site of implantation, and volume/number of tumour cells inoculated. The lack of a standardised approach also complicates comparison while potentiating different experimental outcomes. Finally, the development time and extent of diffuse, infiltrative growth, and metastasis make these models difficult to compare to human disease progression. Because metastases at diagnosis is a relatively rare occurrence in DMG patients, optimising the cell implantation procedure in a standardised manner could better mimic tumour growth progression in vivo, with a greater correlation with human disease progression.

We postulated that using Matrigel instead of phosphate-buffered saline (PBS) as a cell suspension matrix for tumour cell inoculation in preclinical models would prevent premature cell dissemination. Local confinement of the tumour is particularly relevant for locoregional treatment paradigms, such as convection-enhanced and focused ultrasound-mediated drug delivery to the brainstem in preclinical research. The aim of this study is to provide a literature overview of the HSJD-DIPG-007 DMG PDX model, to extract common features, and to investigate the impact of dissimilarities. For the latter, the presented work focuses on the comparison of the extent of infiltrative and metastatic growth patterns of the model with cells inoculated with either PBS or Matrigel as the suspension substrate in athymic nude mice and the relevance of the time delay between inoculation and the onset of therapy.

## 2. Materials and Methods

### 2.1. Literature Search

A literature search was performed to identify publications using the HSJD-DIPG-007 cell line to create an overview of preclinical DMG tumour models using this cell line without any exclusion criteria. Upon study inclusion, data were classified based on (1) animal host and age, (2) location of injection, (3) injected volume and cell concentration, (4) cell suspension matrix, and (5) treatment and follow-up. The age of the mice was categorised based on their postnatal (≤3 weeks), adolescent (3–9 weeks), and adult (>9 weeks) phase [33].

### 2.2. Animals

All experiments were conducted on 6–8-week-old male athymic nude Foxn1^-/-^ mice (Code 069, Envigo, Horst, The Netherlands) in accordance with guidelines of the Dutch Ethical Committee and the Animal Welfare Body of Utrecht University (AVD3990020209445, approval date: 11/02/2020). A total of 34 mice were used for the study, consisting of 15 for DMG PDX /tumour growth and survival validation and 19 that were sacrificed at designated timepoints for histological analysis. Mice were housed under specific pathogen-free conditions in separately ventilated cages, at up to four animals/cage, and allowed to acclimatise for 2 weeks before experimental procedures. Mice were kept on regular laboratory food and water ad libitum, with a fixed 12 h light/dark cycle in accordance with ARRIVE guidelines [2]. Measurable outcomes in PDX models of DMG are not influenced by gender, and as such gender dimension was not relevant for this study [34]. A detailed description of housing conditions of animals is available as Appendix A.

### 2.3. Cells

HSJD-DIPG-007 cells are patient-derived from the autopsy of a pontine tumour in a 9-year-old male, kindly provided by Dr. Ángel Montero Carcaboso (Sant Joan de Déu Barcelona Hospital). Genetic information is archived in the Cellosaurus Database (CVCL_VU70, www.cellosaurus.org). Cells were grown and maintained in 1:1 Neurobasal-A and Advanced DMEM/F-12 medium containing 10 mM HEPES buffer, 1× MEM nonessential amino acids, 1% GlutaMAX, 1 mM Sodium pyruvate, 1× B-27 minus vitamin-A (ThermoFisher, Waltham, MA, USA), 10 ng/mL PDGF-AA, 10 ng/mL PDGF-BB, 20 ng/mL bFGF, 20 ng/mL EGF (Peprotech, London, UK), 2 µg/mL heparin (Leo Pharma, Amsterdam, The Netherlands), and 1 mg/mL primocin (InvivoGen, San Diego, CA, USA). Medium was refreshed every 3–4 days. Single-cell suspensions were obtained using Accutase (ThermoFisher, Waltham, MA, USA). Cells were cultured at 37 °C, 5% CO_2_, and 95% humidity. For in vivo tumour growth monitoring by bioluminescence imaging (BLI), HSJD-DIPG-007 cells were transduced to express firefly luciferase as previously described [35]. Following infection, eGFP-lucF-gene-positive HSJD-DIPG-007 cells were selected using a Sony SH800 Cell Sorter (Sony, Tokyo, Japan). Before cell implantation, HSJD-DIPG-007 cells were suspended in 1× PBS (pH 7.4) or Matrigel (50% *v*/*v*, in PBS, Corning, Corning, NY, USA) and kept on ice until used.

### 2.4. Drugs

Pre- and postsurgical analgesia was managed with 67 µg/mL carprofen (Faculty of Veterinary Medicine pharmacy, Utrecht, The Netherlands) per os (p.o.) in drinking water with an additional subcutaneous (s.c.) injection of 5 mg/kg before surgery. Further pain suppression was performed by s.c. injection of 0.5% lidocaine (B. Braun, Melsungen, Germany) during surgery. Anaesthesia was maintained with isoflurane (Zoetis, Capelle aan den IJssel, The Netherlands), mixed with air, 3% induction, 1.8% maintenance. BLI signal of engrafted cells was monitored by intraperitoneal (i.p.) injection of 150 mg/kg D-luciferin (Cayman Chemical, Uden, The Netherlands) in PBS. Euthanasia was performed via i.p. injection of a mix of 7.14 mg/mL ketamine (Alfasan, Woerden, The Netherlands) and 0.714 mg/mL sedazine (AST Farma, Oudewater, The Netherlands) in PBS.

### 2.5. Tumour Cell Implantation

Twenty-four hours before and after orthotopic intracranial injection with eGFP-lucF-gene-positive HSJD-DIPG-007 cells, mice received carprofen p.o. in drinking water. Thirty minutes before surgery, carprofen was administrated s.c. for local pain management. After anaesthesia with isoflurane, mice were fixed on a stereotactic frame with bite and ear bars. Eye cream was applied to prevent eye damage, while the mice were kept warm during the procedure. After incision of the skin, a drop of lidocaine was added before removal of the facia on the skull. Using a high-speed drill, a burr hole was made in the skull 0.8 mm posterior and 1.0 mm lateral to the lambda. At a depth of 4.5 mm in the pontine region, a total of 5 × 10^5^ HSJD-DIPG-007 cells suspended in 4.3 µL of PBS or Matrigel were injected at a rate of 2 µL/min using a 5 µL Hamilton syringe fitted with a 26-gauge needle. After injection, the needle remained in place for 7 min before being slowly retracted to prevent cell accumulation in the needle tract. Wound closure was performed by applying topical skin adhesive Histoacryl (B. Braun, Melsungen, Germany) before placing the mice under a heating lamp until awake. Possible signs of stress and postoperative complications (lack of food/water intake, antisocial behaviour, and motor deficits) were carefully monitored.

### 2.6. Tumour Growth Assessment with Bioluminescence

Mice were weighed three times a week, while their tumour growth was monitored twice a week by measuring the BLI signal of engrafted eGFP-lucF-gene-positive HSJD-DIPG-007 cells using the MILABS U-OI camera (MILABS, Houten, The Netherlands). For signal measurement, mice were anaesthetised with isoflurane and injected 5 min later with D-luciferin before positioning in the camera. BLI images were taken under anaesthesia from 5 to 30 min after D-luciferin injection with a 60 s exposure. Signal intensity was quantified within the region of interest (ROI) of the whole animal head by using ImageJ software (v1.53t, National Institute of Health, Bethesda, USA) [36]. Mice were sacrificed with ketamine/sedazine after reaching their scientific or humane endpoints. Humane endpoints were defined based on 20% weight loss from cell implantation, 15% weight loss within two days, or development of neurological deficiencies, such as circling, hyperexcitability, convulsions, or ataxia.

### 2.7. Histological Analysis

Histopathological elements, tumour size, location, and proliferation were determined by human vimentin and haematoxylin and eosin (H & E) staining. After euthanasia, mice were transcardially perfused with PBS followed by 10% formalin, after which the brain was excised and postfixed in 10% formalin for 48 h before paraffin embedding. Sagittal sections of 4µm were made using a microtome (Leica Biosystems, Wetzlar, Germany) and mounted on Superfrost^®^ Plus microscope slides. Before staining, sections were deparaffinized and subjected to antigen retrieval with sodium citrate buffer (10 mM, pH 6, 95–100 °C, and 30 min). Endogenous peroxidase activity was reduced by incubation in 3% hydrogen peroxidase for 20 min, after which sections were washed twice with deionized water and once with 1× Tris-buffered saline containing 0.1% Tween (TBST). Sections were blocked for 1 h at room temperature with antibody diluent clear (VWRKBD09-125, VWR, Radnor, USA) before overnight incubation at 4 °C with rabbit antihuman vimentin [SP20] (1:5–1:8, ab27608, Abcam, Cambridge, UK) followed by washing with TBST. Sections were then incubated for 2 h at room temperature with biotinylated affinity-purified goat antirabbit secondary antibody (1:500, BA-1000, and IgG (H + L), Vector Laboratories, Newark, NJ, USA) before washing with TBST. VECTASTAIN^®^ Elite ABC-HRP Peroxidase (PK-6100, Vector Laboratories, Newark, USA) was applied for 1 h at room temperature followed by a 3–4 min incubation in 3,3′-diaminobenzidine (DAB, K346711-2, Agilent Dako, Amstelveen, The Netherlands) before counterstaining with haematoxylin (Epredia, Breda, The Netherlands).

### 2.8. Data and Statistical Analysis

Weight and tumour growth measured by BLI signal were analysed using an independent t-test. Survival was analysed using a Kaplan–Meier plot and Log-rank test. Metastases formation in olfactory bulb and spinal cord were analysed by a nonparametric Kolmogorov–Smirnov test to compare cumulative distributions. A *p*-value of ≤0.05 was considered statistically significant. Statistical analyses were performed using GraphPad Prism (v9, GraphPad Software, LLC, Boston, MA, USA). Photographic and electronic images were obtained on a Leica DMi8 and processed using Adobe Photoshop 21 (Adobe Inc., San Jose, CA, USA).

## 3. Results

### 3.1. HSJD-DIPG-007 PDX Model in the Literature

A total of 20 articles were published between 2016 and 2022 using the HSJD-DIPG-007 cell line for establishing a DMG PDX mouse model [5,25,31,32,37,38,39,40,41,42,43,44,45,46,47,48,49,50,51,52]. An overview of these studies is given in Table 1. For the orthotopic generation of DMG, 65% of the studies described injection in the pontine/brainstem region, 20% in the 4th ventricle, and 15% in a combination of both 4th ventricle/pons. In 75% of the studies, adolescent mice were used for establishing the tumour model, 15% used early postnatal mice, and 10% did not define the age. Athymic nude, nude BALB/c, NOD-SCID, and NOD-SCID gamma (NSG) nude mice were used as host animals in 30%, 20%, 30%, and 15% of the cases, respectively, with one study (representing 5%) using an athymic nude rat. The injection volume ranged between 1 µL and 5 µL, with 45% of the cases injecting 5 × 10^5^ HSJD-DIPG-007 cells. Only one study used 7.5 × 10^5^ cells suspended in 7.5 µL for establishing the PDX model using athymic nude rat as the host. PBS, Matrigel, or medium were used as suspension matrices in 20%, 40%, and 10% of the studies reported, respectively. In 20% of the studies, an undefined suspension matrix was used, while the remaining 10% used combinations or other matrices. The treatment applications ranged from day 0 up to day 80 after cell inoculation. Despite the high variety of treatments performed in these studies, prolonged survival or delayed tumour growth was observed in 82% of cases reporting treatment outcomes.

### 3.2. Well-Being and Weight Profiles upon Implantation Procedure

Orthotopic injections of HSJD-DIPG-007 cells suspended in PBS (n = 9) or Matrigel (n = 6) did not give rise to deleterious neurological complications following implantation. The time frame of the surgery and anaesthesia affected the wakefulness of the mice afterwards, where extended procedures resulted in lengthier recovery times until the mice were fully active and mobile (observation), even though mouse core temperature was monitored and maintained throughout the procedure. Following cell implantation, mice initially lost weight, but gained on average 16% of their initial weight by day 37 for PBS and 15% by day 30 for Matrigel-injected mice. No significant differences in overall weight gain or loss were measured between the PBS and Matrigel groups (Figure 1). An early and aggressive tumour onset can explain the severe weight loss in one PBS mouse (Appendix A).

### 3.3. Survival and Tumour Growth Using PBS or Matrigel as Suspension Matrices

Despite different suspension matrices being used, no significant differences in survival between the PBS and Matrigel groups were observed. Mice with PBS or Matrigel survived up to 90 and 100 days and with a median overall survival of 70 and 75 days, respectively (Figure 2A). PBS mice were sacrificed in 2/9 cases based on neurological symptoms of motor functions, such as tremors and paralysis, 6/9 based on weight loss, and 1/9 for both conditions. Matrigel mice were sacrificed in 2/6 cases based on neurological symptoms, 3/6 based on weight loss, and in 1/6 case, the animal passed away during BLI. The BLI signal confirmed successful cell implantation in all animals of both treatment groups. Steady exponential tumour growth was observed up to day 30 post implantation, after which the growth increase exceeded around day 40 1.8 AU/day for both (Figure 2B,C and Appendix A). The increased exponential growth could be indicative of locoregional metastasis formation with the tumour spreading outside the injection location of the pons. No significant differences in tumour growth were observed between the PBS and Matrigel groups.

### 3.4. Metastases Occurrence in Olfactory Bulb and Spinal Cord

Because DMGs are tumours beginning in the pontine region and spreading on mid-disease in the majority of cases to adjacent areas, the HSJD-DIPG-007 PDX model should preferably recapitulate this growth pattern, in particular for the evaluation of locoregional treatment at the initial stage of disease [53]. Based on the BLI signal, the first onset of metastases in the frontal lobe (olfactory bulb) was observed at day 26 after inoculation in the PBS group and at day 44 after inoculation in the Matrigel group. A median metastasis-free survival (MMFS) in the olfactory bulb of 33 vs. 58 days was found for PBS and Matrigel mice, with a significant difference between the groups (Figure 3A). Metastatic formations in the spinal cord were first observed at day 37 post inoculation in the PBS group and at day 44 post inoculation in the Matrigel group, with an MMFS of 47 and 68 days, respectively (Figure 3B). At time of death, two of the nine mice in the PBS group had not developed metastases, while only one had metastasis in the olfactory bulb. Of the six Matrigel mice, two did not develop metastases, and one developed an olfactory bulb metastasis.

Mice with cells suspended in PBS showed local growth up to day 31, with subsequent locoregional progression as well as metastatic formations in the mid cerebrum/lateral ventricle, with eventual spreading into the cerebellum and olfactory bulb. Mice with cells suspended in Matrigel showed local growth up to day 31 and the presence of tumour cells in the lateral ventricles, with delayed locoregional progression and distal striatal infiltration with inevitable invasion of the whole brain at day 55 (Figure 4). No histopathological or morphological differences were observed in the mice of both groups by H & E staining (Appendix A). Antihuman vimentin staining confirmed the local injection of HSJD-DIPG-007 in the pontine region of the mice and showcased that contamination of the cerebrospinal fluid (CSF) and dissemination to other brain structures in proximity to the injection site through the perivascular system (PVS) can occur (Figure 5).

A comparative histopathological analysis of clinical autopsy-derived DMG with the orthotopic E98 DIPG mouse model was previously performed by Caretti and colleagues [53]. To determine the clinical relevance of the HSJD-DIPG-007 PDX model, we used the histology panel of DMG patient tissue of Carreti et al. for a comparative assessment of disease progression (Figure 6). Perivascular tumour dissemination in the HSJD-DIPG-007 PDX model was seen to be like that observed in the DMG patient (Figure 6C,D). Similarities were also observed in brain parenchyma invasion in the HSJD-DIPG-007 model and clinical DMG (Figure 6G,H), as well as vascular proliferation (Figure 6K,L).

## 4. Discussion

DMG is an invasive paediatric brain tumour with a high mortality rate, and because of the location and infiltrative nature of the disease, radiotherapy is the only effective palliative treatment option currently available [10,17]. As DMG rarely develops naturally in animals, PDX animal models are important for preclinical in vivo therapy efficacy validation. However, due to translational complexities between preclinical research and clinical applicability, there is a demand for PDX models emulating human disease progression. Metastases and immediate organ-specific proliferation infrequently occurs in patients in early stages of disease progression but can be seen in late/end stages of DMG [19]. Ideally DMG models would reproduce this form of early disease progression, which is observed in most of the patients as an initial diffuse local tumour proliferation in the pontine area, with subsequent expansion through the medulla, the cerebellum, and the thalamic areas. Because DMG patients suffer from a rapid progression of disease in the vital pontine area leading to a poor prognosis, late-stage disease beyond this point is rarely observed.

The HSJD-DIPG-007 cell line, derived from the autopsy of a 6-year-old, is widely used for establishing DMG PDX models, but the high heterogeneity among protocols indicates that a universal procedure is yet to be developed. With the aim of facilitating a standardised inoculation method, this study investigated and optimised the growth pattern of the HSJD-DIPG-007 PDX model for local or metastatic phenotype treatment based on two different suspension matrices.

Studies in other cancer models, such as pancreatic cancer, have shown the importance of the suspension matrix when tumour cells are injected locally and the issues, such as leakage, low tumour formation, and development of metastases, that can arise [54]. The local introduction of cells into the brain is a delicate matter, requiring precision in location, as well as in the injection procedure to avoid positive and negative pressure build-up that could dissipate the cells in an unfavourable manner. A possible alternative to common suspension matrices, such as growth medium and PBS, could be the basement membrane Matrigel, due to its composition resembling the extracellular matrix of many tissues, as well as its favourable viscoelastic properties where it remains liquid at low temperatures but polymerises to a dense matrix at temperatures above 10 °C [55]. No standardised procedure in establishing the HSJD-DIPG-007 PDX model could be discerned from the studies outlined in Table 1. Protocols differed considerably in all the reported parameters, as well as in the stated level of detail provided, with the most noteworthy differences being in the suspension matrix used, day at which treatment was initiated, and treatment modality or efficacy. In our study comparing PBS and Matrigel, we selected to use animals at 6–8 weeks of age for inoculation of the HSJD-DIPG-007 cell line, corresponding to the age range used by 50% of the studies reported in Table 1 and commonly used for in vivo studies. To ensure adequate cell grafting, we also opted for 5 × 10^5^ total cell inoculation for both the PBS and Matrigel suspension groups.

The initial observation made in comparing the PBS and Matrigel groups was in weight stability following HSJD-DIPG-007 cell implantation. Substantial fluctuations, including rapid gains/losses in a short period of time are reliable indications of health in in vivo animal models. In our study, no significant differences in weight were observed between the PBS and Matrigel groups. The weight gained in the first 4–5 weeks after cell implantation could be due to the young 6–8-week age of the mice, in which they were still in their adolescent and body growth phase [33].

As DMG progresses quite rapidly in children, symptoms are typically not evident for 4 to 6 weeks before diagnosis [56]. Patients present to the clinic when disease progression is relatively advanced, with a triad of symptoms consisting of cranial neuropathy, long tract signs, and cerebellar signs [57]. By this stage, DMG may have been developing for 12 months or more. In our study, mice in both groups were asymptomatic and gained weight for 30 days, as seen in Figure 1, after which tumour presence could be verified and followed by BLI, and weight loss began to occur. The subsequent weight loss could be attributed to the progression of the tumour in the pontine region, the consequence of which could be diminished appetite. Figure 2 shows how the BLI signal intensified rapidly post day 30, with a strong signal being observed in brain regions outside the graft area, suggesting the presence of metastatic formations. However, this increase in the BLI signal did not correspond with overall survival, as no significant differences were observed irrespective of whether cells were inoculated using PBS or Matrigel. Both groups also had comparable tumour growth rates during the steady growth phase of the first 4 weeks post inoculation as well as in the exponential growth phase thereafter, further supporting the similar survival times observed and confirming that the suspension matrix on its own does not influence local tumour growth or survival.

Strikingly, the analysis of the BLI signal did show differences between the PBS and Matrigel groups in terms of metastatic formations within the olfactory bulbs, as seen in Figure 3, suggesting that Matrigel does significantly delay the onset of metastases by an average of approximately 3 weeks. Delays in the spinal cord were also observed in the Matrigel group, and even though these were not found to be significant, the suggestion that Matrigel influences metastasis formation is present. The polymerisation of the cell-loaded Matrigel upon injection into the pons could have contributed to reduced cellular leakage into the brain parenchyma or into the needle tract produced during the inoculation procedure, without altering the tumour growth rate. The observation that the use of one suspension matrix significantly delays metastases when compared to another further emphasises the need for a standardised protocol in establishing DMG PDX models through orthotopic injection of cells into the pons. This was confirmed, as seen in Figure 4, through antihuman vimentin staining of HSJD-DIPG-007 cell-inoculated mouse brains at various stages of tumour development. The staining confirmed that the pons was accurately targeted and that the cells successfully engrafted and were able to induce local tumour formation. Tumour growth rapidly progressed with time, while advanced metastatic formations were observed by 31 days within the midbrain and by 55 days within the olfactory bulbs of the PBS suspension group and not in the Matrigel group.

The PVS and cerebrospinal fluid hydrodynamics are important factors that must also be considered, especially when DMG PDX models are established. It has previously been shown that connections between the CSF and nasal lymphatic vessels in mammals, including humans and rodents, share common characteristics [58]. Metastases can initially be seen in the location of the lateral ventricle, followed by formations in the olfactory bulbs, which coincides with the direction of the CSF flow in both humans and rodents. In humans, CFS circulates in a caudal-directed manner through the ventricles to the subarachnoid space, resulting in an exchange of various substances in a to-and-from manner between the CSF and interstitial compartments [59].

A proportion of the CSF drains into the cribriform place, while the rest is recycled into the brain parenchyma through perivascular spaces surrounding blood vessels. Perivascular space connections penetrating deep into the brainstem and 4th ventricle have also been observed. New PVS connections between ventricles and different parts of the brain parenchyma have been revealed, suggesting a possible role for the ventricles as a source or sink for solutes in the brain [60].

These observations further demonstrate that Matrigel, as a suspension matrix, is more favourable in supporting local tumour growth at the site of inoculation and delays the onset of metastases, especially to the olfactory bulbs, in a significant manner, further supporting its use as a more suitable suspension matrix than PBS. It may be that Matrigel supresses perivascular proliferation of inoculated cells, resulting in a model of disease progression that more closely resembles that seen in patients as described by Caretti and colleagues [19].

Although Matrigel delays metastases overall, the inoculation procedure itself is not infallible. When injecting a substance 4–5 mm deep within the mouse brain, the needle passes through several structures and does cause a degree of disruption to adjacent tissues, while also disturbing the CSF present in the brain. As shown in Figure 5, rogue cells can be seen already circulating within the brain outside the pontine region immediately following inoculation. The circulating CSF could potentially distribute these cells through the lateral ventricles and on to the olfactory bulbs, which also act as a CSF sink and outflow to the nasal lymphatics [60], where they can give rise to metastatic formations developing very early following initial inoculation. Therefore, it is imperative that any residual cells that may remain on the outside of the needle while filling with cell-containing suspension matrix be removed thoroughly before injection into the brain.

When the observations of tumour volume and BLI signal of Figure 3 are compared with the immunohistochemical images of the tumour progression in Figure 4, we can see that although the BLI signal is not detected before 40 days post inoculation of the HSJD-DIPG-007 cells, tumour progression with extensive infiltration of the brain parenchyma by tumour cells is already present by day 21. This suggests that the BLI signal alone is not reliable in determining early and local tumour formations and thus should not be used as a measure to determine the onset of treatment as tumour size and burden, including the presence of metastases could be underestimated. Such an underestimation could render treatment regimens unsuccessful because of a too large tumour burden rather than treatment inefficacy, leading to false negatives and the ultimate rejection of suitable drug or treatment candidates. From the immunohistological data obtained, in addition to using Matrigel as a cell suspension, we would suggest that treatment initiation be performed between 7 and 14 days post cell inoculation. This timeframe would allow for cells to engraft and tumour formation to occur to a point where the burden is not too high to render treatment ineffective and not too low to result in false positives. It is noteworthy that of the studies listed in Table 1, only 1/5 initiated treatment within this timeline, while the majority started therapy three or more weeks following cell implantation. For locoregional therapy approaches, treating within two weeks would also ensure that the entire tumour within the pons is targeted, and not later-occurring metastatic formations within other brain regions, which are missed, especially in the distant olfactory bulbs.

In summary, the HSJD-DIPG-007 PDX mouse model is one that has gained interest in DMG research as it does resemble human disease progression in a clinically relevant manner, as Figure 6 shows. Vital elements, such as perivascular tumour dissemination, invasion of the parenchyma, and vascular proliferation, are well emulated in the HSJD-DIPG-007 PDX model. As Caretti [53] showed in both the clinical and E98 DMG tumours and observations in the HSJD-DIPG-007 model used in this study, perivascular migration appears to be a route by which invasion of the brain parenchyma can occur by tumour cells located in the subarachnoid space. However, for this model to be optimally utilised in preclinical research, standardisation of its establishment needs to be achieved.

Based on our results, we propose a standardised method of using Matrigel as a suspension matrix to inoculate cells within the pons to delay metastases to other brain regions. We also suggest treatment initiation be within 1–2 weeks of grafting to ensure an adequate but not overbearing tumour burden for assessment of treatment strategies. Further standardisation of this model assessing the animal host used, total cells inoculated, injection volume, and graft location is needed so that a reliable and reproducible model that recapitulates the histological characteristics of DMG can be established.

## Figures and Tables

**Figure 1 biomedicines-11-00527-f001:**
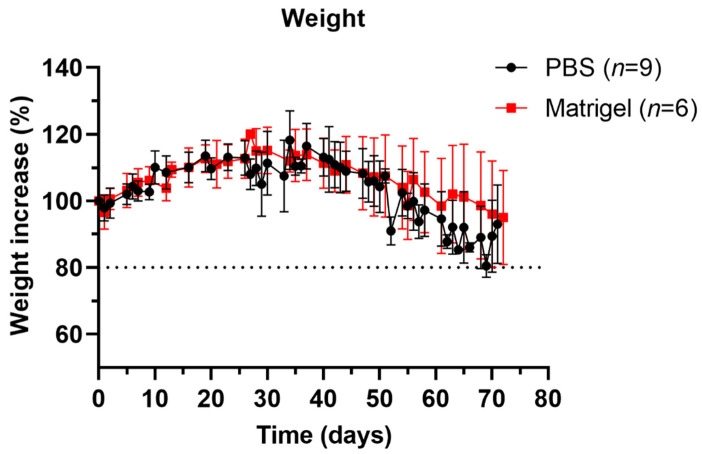
Weight profiles of mice inoculated with HSJD-DIPG-007 cells suspended either in PBS or Matrigel. Changes in weight after intracranial implantation of HSJD-DIPG-007 cells suspended in either PBS or Matrigel were monitored. Weight increased consistently up to day 37 for PBS and day 30 for Matrigel suspension groups, after which weight loss set in, lasting until terminal endpoint. No significant differences between PBS and Matrigel groups were observed. Dotted line represents the weight threshold of the humane endpoint. Data points are expressed as mean weight ± SD.

**Figure 2 biomedicines-11-00527-f002:**
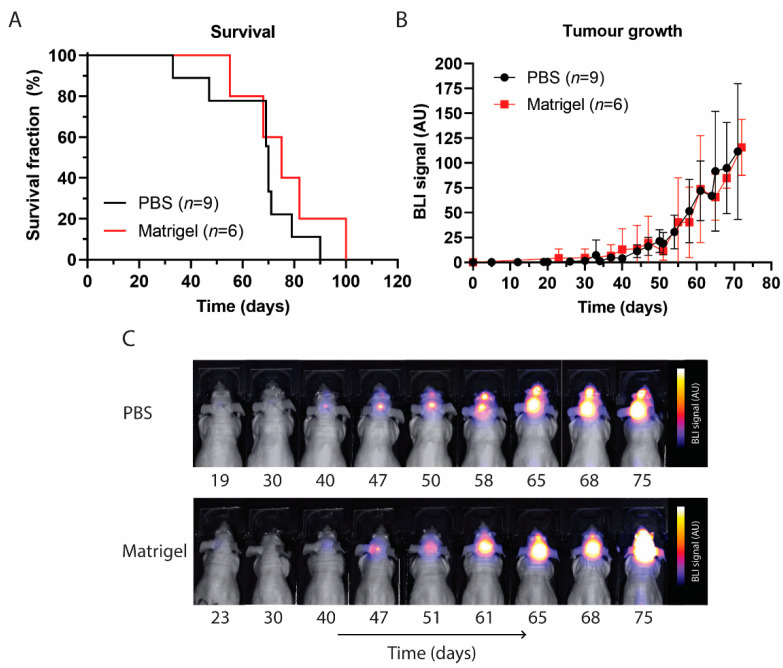
Survival and tumour growth following inoculation with HSJD-DIPG-007 cells suspended in PBS or Matrigel. (**A**) Kaplan–Meier curve showing survival following cell inoculation and tumour progression. No significant difference between PBS and Matrigel suspension groups was observed. (**B**) Tumour volume over time in both PBS and Matrigel suspension groups, showing that tumour growth was comparable up to 75 days. Data points are expressed as mean signal intensity ± SD. (**C**) BLI signal showing tumour growth over time. Tumour development within the pontine region, as well as metastatic development in the olfactory bulb region, can be seen in both PBS and Matrigel groups, progressing with time. AU = arbitrary unit.

**Figure 3 biomedicines-11-00527-f003:**
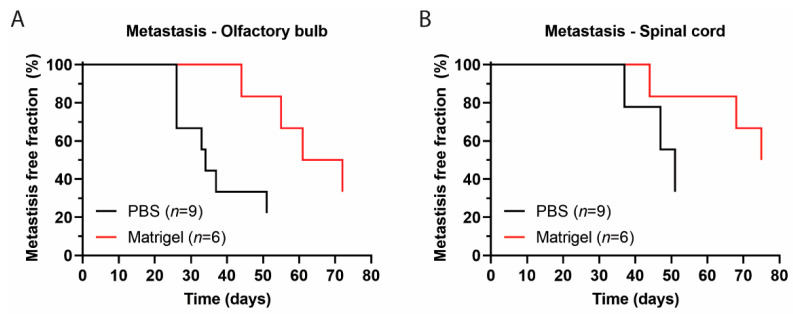
Distant metastatic formations over time monitored through BLI signal following HSJD-DIPG-007 cell inoculation into the pontine region. (**A**) Metastasis in the olfactory bulb in PBS and Matrigel suspension groups with a median onset of 33 and 53 days, respectively. A significant difference between the two groups was found (*p* < 0.05). (**B**) Metastasis in the spinal cord in PBS and Matrigel suspension groups with a median onset of 47 and 68 days, respectively, but without a significant difference (*p* > 0.05).

**Figure 4 biomedicines-11-00527-f004:**
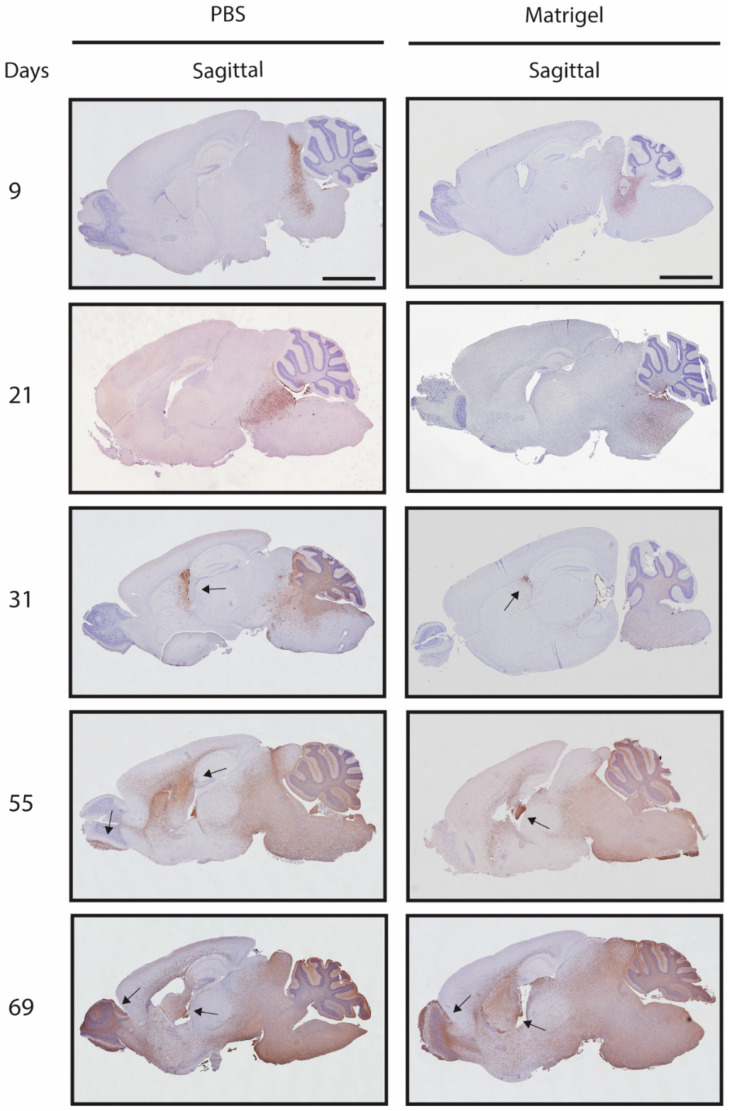
Antihuman vimentin staining of mouse brains showing tumour progression over time following inoculation with HSJD-DIPG-007 cells suspended in PBS or Matrigel. Tumour progression over time can be seen in both PBS and Matrigel groups through the accumulation and spread of human vimentin-positive cells (brown staining) within the pons and other, more distant brain regions. Metastases can be observed from day 31 in both PBS and Matrigel suspension groups (black arrows). Whole brain invasion of tumour cells can be observed at day 55 in both groups. Mouse brains in both groups were counterstained with haematoxylin. *n* = 9 for PBS group and *n* = 10 for Matrigel group. Scale bar = 2 mm.

**Figure 5 biomedicines-11-00527-f005:**
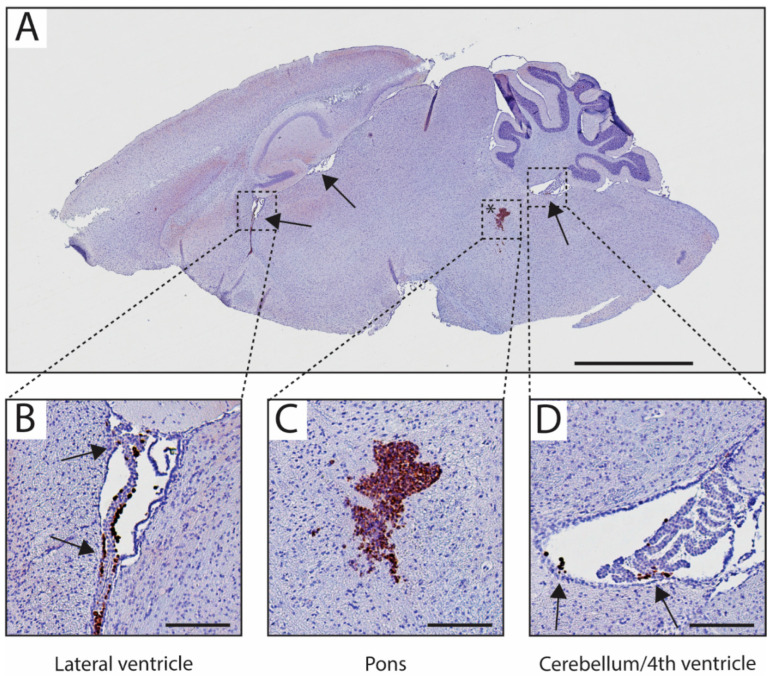
HSJD-DIPG-007 cells within the pons and disseminated throughout the brain immediately following inoculation. (**A**) Matrigel-suspended HSJD-DIPG-007 cells within the pons area (asterisk) as well as in distant brain structures (arrows) at time zero, identified via antihuman vimentin staining. Magnification of HSJD-DIPG-007 cells present in the choroid plexus of the lateral ventricle (**B**), pons (**C**), and choroid plexus of the cerebellum/4th ventricle (**D**). Counterstaining is with haematoxylin. Scale bar = 2 mm for **A**, 100 µm for (**B**–**D**).

**Figure 6 biomedicines-11-00527-f006:**
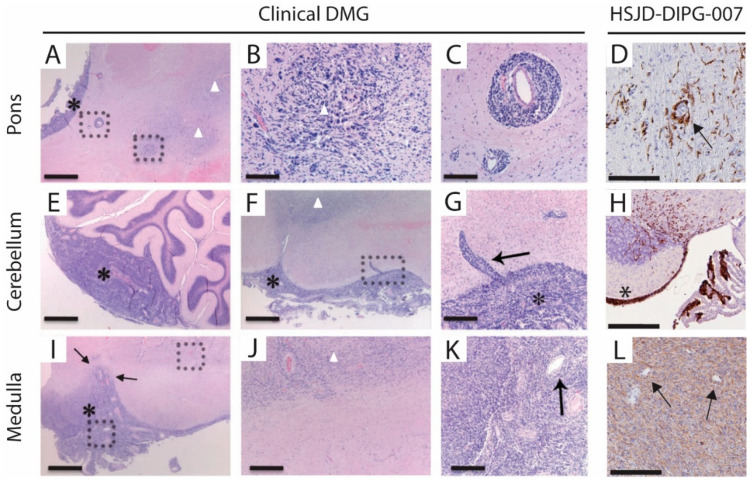
Comparative assessment of clinical DMG and HSJD-DIPG-007 PDX model histopathology. The comparative clinical DMG panel (panes **A**–**C**,**E**–**G**,**I**–**K**) was adapted and reproduced with permission from Caretti et al. [53]. (**A**,**E**,**F**,**I**) H & E staining of clinical DMG at the pons, cerebellum, and medulla. (**B**,**C**,**G**,**J**,**K**) Magnifications of H & E staining indicated by the dotted squares. (**D**,**H**,**L**) Antihuman vimentin staining of HSJD-DIPG-007 PDX model at the pons, cerebellum, and medulla. (**A**,**E**–**I**) Asterisks indicate leptomeningeal growth. (**D**,**G**) Arrows indicate perivascular growth, and (**I**,**K**,**L**) represent blood vessels in dense tumour areas. Scale bars = 250 µm (**A**,**F**,**I**), 62.5 µm (**B**,**C**,**G**), 125 µm (**J**,**K**), 500 µm (**E**), and 50 µm (**D**,**H**,**L**).

**Table 1 biomedicines-11-00527-t001:** Summary of studies using HSJD-DIPG-007 for establishing a DMG PDX model from 2016 to 2022.

Animal Host	Age (Weeks)	Location	Total Cells Inoculated	Volume	Suspension Matrix	Days Before Treatment	Treatment Strategy	Treatment Efficiency	Reference
**Athymic nude**	n.d.	Brainstem	5 × 10^5^	n.d.	n.d.	21	RG7388	Enhanced survival	[5]
**NOD-SCID**	7	Pons	3 × 10^5^	2 µL	Matrigel	28	Panobinostat	No	[25]
**NOD-SCID**	5	Pons	2 × 10^5^	5 µL	Matrigel	None	None	Not assessed	[31]
**Athymic nude**	6	Pons	5 × 10^5^	5 µL	PBS	37	Doxorubicin and FUS	No	[32]
**Athymic nude**	n.d.	Pons	5 × 10^5^	5 µL	n.d.	14	BGB324, Panobinostat and CED	Enhanced survival combined with CED	[37]
**NOD-SCID**	6–8	4th Ventricle	5 × 10^5^	4 µL	Matrigel	28	OKN-007 and LDN-193189	Reduced cellular activity	[38]
**NOD-SCID gamma (NSG)**	8–10	Pons	4 × 10^5^	2 µL	Medium:Matrigel (1:1)	21	2-DG and IDH1 inhibitor	Enhanced survival and decreased growth	[39]
**Athymic nude rat**	4	4th Ventricle	7.5 × 10^5^	7.5 µL	n.d.	28	SN-38	Not assessed	[40]
**Nude BALB/c**	8	4th Ventricle/Pons	2 × 10^5^	3 µL	n.d.	21	DCA, Metformin and RT	Enhanced survival combined with RT	[41]
**NOD-SCID**	7–8	4th Ventricle/Pons	2 × 10^5^	2 µL	Matrigel	28	CBL0137 and Panobinostat	Enhanced survival in combination	[42]
**Nude BALB/c**	6	Pons	1 × 10^4^	1 µL	HBSS	21	CRAd.S.pK7	Enhanced survival	[43]
**NOD-SCID gamma (NSG)**	5–7	Pons	1 × 10^5^	2 µL	Serum free media	80	ALDH+/− and GDC-0084	Enhanced survival	[44]
**Nude BALB/c**	5–7	Brainstem	2 × 10^5^	2 µL	Matrigel	30	DMFO & AMXT	Enhanced survival and decreased growth	[45]
**Athymic nude**	3	4th Ventricle	5 × 10^5^	5 µL	Matrigel	25	Vandetanib and Everolimus	Enhanced survival in combination	[46]
**Athymic nude**	7–9	Striatum/Pons	5 × 10^5^	5 µL	PBS	75	Bevacizumab	Not assessed	[47]
**Athymic nude**	6–8	Pons	5 × 10^5^	5 µL	PBS	7–8	Doxorubicin and CED	No	[48]
**NOD-SCID**	4–5	4th Ventricle	5 × 10^5^	2 µL	PBS	0	HSV1716	Reduced cellular growth	[49]
**Nude BALB/c**	5–6	4th Ventricle/Pons	2 × 10^5^	2 µL	Matrigel	28–35	Temozolomide and RT	RT enhanced survival	[50]
**NOD-SCID gamma (NSG)**	0–2 days	Brainstem	1 × 10^3^	n.d.	Tumour stem medium	0	GSK2830371	Enhanced survival	[51]
**NOD-SCID**	3	Pons	5 × 10^5^	5 µL	Matrigel	21–28	LDN-193189 and LDN-214117	Enhanced survival	[52]

n.d.: Not defined.

## Data Availability

No new data sets were created or analysed in this study therefore data sharing is not applicable for this article.

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
