# Peer review of "Towards Standardisation of a Diffuse Midline Glioma Patient-Derived Xenograft Mouse Model Based on Suspension Matrices for Preclinical Research"

_biomedicines, 2023, doi:10.3390/biomedicines11020527_

Round 1
Reviewer 1 Report
Towards standardisation of a diffuse midline glioma patient- 2 derived xenograft mouse model based on suspension matrices 3 for preclinical research IS EXCELLENT Manuscript. Will be useful for Glioblastoma Research
Reviewer 2 Report
The manuscript is related to preclinical model development for midline glioma and used materials for inoculation of cells to the mice. They have been used the suspansion of cells to intracranium by using matrigel or PBS. Then they have evaluted the tumor growth, survival and metastases. They have only showed the delayed metastases by using matrigel treatment compared to PBS treated group. The issue of manuscript is very simple for development of patient derived xenograft models for gliomas. Also, the manuscript does not give us any new data for the application of PDX models for gliomas. So that, it can not acceptable for the publication.
Reviewer 3 Report
The paper is really reliably prepared, a this is hard to show weak points. The only thing which is lacking for me is a scale bar showing the relationship between BLI intensity and colour in Figure 2.
